# Zero-Direction Probing:
# A Linear-Algebraic Framework for Deep Analysis of Large-Language-Model Drift

## Abstract

We present **Zero-Direction Probing** (ZDP), a theoretical framework that characterises model drift from *null* directions of transformer activations, requiring no task labels or output evaluations. Under explicit assumptions (A1–A6), We prove: (i) the *Variance–Leak Theorem* (Thm. 1), (ii) *Fisher Null-Conservation* (Thm. 2), showing that second-order KL curvature is confined to the base image space up to a residual controlled by Fisher leakage into null directions, (iii) a *Rank–Leak* bound for low-rank updates (Thm. 4), and (iv) a logarithmic-regret guarantee for online null-space trackers (Thm. 3). We further derive a *Spectral Null-Leakage* (SNL) metric with a non-asymptotic Laurent–Massart tail bound and an MP-edge–style concentration inequality, providing a-priori thresholds for drift under a Gaussian null model. Together, these results establish that "listening to silence"—monitoring the right/left null spaces of layer activations and their Fisher geometry—yields concrete, testable guarantees on representational change. The manuscript is intentionally theory-only; empirical validation and benchmarking are deferred to companion work.

## 1 Introduction

Large language models (LLMs) are routinely adapted after pre-training: supervised fine-tuning, preference optimisation, and domain specialisation all change internal representations. Most drift detectors reason *after the fact* using outputs or high-variance latent directions. In contrast, we study the geometry of *zero-variance* directions—the right/left null spaces of layer activations—and ask:

> *What can be **proven** about representational drift by inspecting only the null spaces of the base model, with no access to labels or outputs?*

Our answer is a theory we call **Zero–Direction Probing** (ZDP). Let $H_\ell \in \mathbb{R}^{n \times d}$ denote the activation matrix at layer $\ell$ for the base model, with right-null basis $V_{0,\ell}$ and left-null basis $U_{0,\ell}$. For a perturbed model $\widehat{H}_\ell = H_\ell + \Delta H_\ell$, we quantify *null leakage* via quadratic forms such as $\|\widehat{H}_\ell V_{0,\ell}\|_F^2$. Intuitively, silent directions in the base model are noise-free: any energy or curvature that appears there is unambiguous evidence of change. Throughout, we use the term "activation matrix" to refer specifically to collections of transformer hidden states (residual-stream representations) produced by large language models.

While large-scale empirical benchmarking is deferred to companion work, we include targeted synthetic sanity checks to confirm that key theoretical predictions are observable in controlled settings.

### 1.1 Setting and scope

The paper is entirely theoretical. We state explicit standing assumptions (A1–A6) on ranks, perturbation size, eigengaps, and noise regularity (Sec. 4). All results concern properties of $H_\ell$ and its null spaces; no task labels, outputs, or downstream metrics are used.

### 1.2 Contributions

1. **Linear-algebraic framework.** We formalise right- and left-null spaces for transformer layers, define null-leakage functionals, and relate them to local Gram and Fisher matrices.

2. **Drift theorems.** (Thm. 1) *Variance–Leak* shows that null-space energy lower-bounds the smallest eigenvalue of the local Gram matrix of the perturbation. (Thm. 2) *Fisher Null-Conservation* proves that the second-order KL contribution arises only from components outside the base image space. (Thm. 4) *Rank–Leak Bound* quantifies when low-rank (LoRA) updates re-occupy silent directions via principal angles.

3. **Spectral metric with a priori thresholds.** We introduce *Spectral Null-Leakage* (SNL) and derive non-asymptotic tails: a Laurent–Massart bound for Frobenius energy and an MP-edge style concentration inequality (Lemma 2), yielding parameter-free thresholds under a Gaussian null.

4. **Online guarantees.** We propose *Online Null-Space Tracker* (ONT) and *Online Null-Aligned LoRA* (ONAL) and prove a *logarithmic regret* bound (Thm. 3) under eigengap and noise assumptions, showing that streaming estimates of the null space incur only $O(\log T)$ cumulative excess leakage.

5. **Conceptual implications.** ZDP cleanly separates covariance geometry (NVL/SNL) from information geometry (Fisher), explains when low-rank adaptation leaks into silent directions, and provides null-hypothesis baselines without empirical calibration.

### 1.3 Limitations and outlook

Results depend on accurate null-space estimation (SVD thresholding) and eigengap conditions; finite-sample effects can perturb projectors. Extending the theory to attention-dependent subspaces and non-Gaussian nulls is future work. The manuscript intentionally omits experiments; empirical validation and benchmarking are deferred to a companion study.

### 1.4 Organisation

Section 4 states assumptions and notation. Section 4.1 proves the Variance–Leak theorem. Section 4.2 develops Fisher Null-Conservation. Section 4.3 derives RMT baselines; Section 4.4 presents online tracking; Section 4.6 proves regret bounds; later subsections cover LoRA rank–leak and SNL.

### 1.5 ZDP in Transformer-Based Language Models

Although the theoretical analysis is agnostic to architecture, the motivating setting throughout is a transformer-based large language model (LLM). In this context, the activation matrix $H_\ell \in \mathbb{R}^{n \times d}$ corresponds to the collection of hidden states at layer $\ell$ produced by a fixed pretrained model when processing a batch of $n$ tokens drawn from prompts or sequences. Each row of $H_\ell$ is the residual-stream representation of a token at layer $\ell$, after attention and MLP blocks and prior to the next residual addition.

Fine-tuning, preference optimization, or low-rank adaptation (e.g., LoRA) induces a perturbed model whose corresponding activation matrix is $H_\ell^{\text{ft}} = H_\ell + \Delta H_\ell$ for the same input tokens. The right null space $\ker(H_\ell)$ therefore represents directions in hidden-state space that are never occupied by the base LLM for the given domain or prompt distribution. Any energy or curvature observed in these directions after adaptation constitutes unambiguous representational change.

All null-space probes introduced in this work (NVL, SNL, FNC, and BINA) operate directly on these transformer hidden states and require no access to outputs, labels, or task-specific metrics.

## 2 Related Work

Prior work on representation analysis and drift in deep networks can be organized along three dimensions: (i) which subspaces are studied (dominant vs. silent), (ii) which signals are used (outputs, losses, or internal

geometry), and (iii) whether results are diagnostic or come with formal guarantees. We review these strands through the specific theoretical gaps they leave open, and position Zero-Direction Probing (ZDP) as a framework that addresses each gap using null-space geometry with explicit guarantees.

## 2.1 Representation geometry and similarity measures

A large literature studies representation drift by comparing dominant activation subspaces. Linear probes (Alain & Bengio, 2017), SVCCA (Raghu et al., 2017), PWCCA (Morcos et al., 2018), and CKA (Kornblith et al., 2019) quantify changes in high-variance directions, as do alignment-based analyses of fine-tuning dynamics (Nguyen et al., 2020; Lu et al., 2025).

These methods intentionally ignore near-zero-variance directions and therefore cannot certify when previously silent directions become occupied. ZDP addresses this limitation by elevating the null space of the base activation matrix to a diagnostic object. The Variance–Leak Theorem (Thm. 1) shows that energy observed in these silent directions lower-bounds the strength of the perturbation itself, yielding a label-free and output-free drift certificate.

## 2.2 Null-space constraints and interventions

Several methods exploit null spaces as training-time constraints. LoRA-Null and related approaches restrict low-rank updates to remain orthogonal to selected subspaces in order to reduce forgetting or interference (Qin et al., 2024; Tang et al., 2025). Knowledge-editing methods similarly impose constrained optimization objectives to localize behavioral changes (He et al., 2025).

While effective in practice, these approaches do not characterize when or how updates re-enter silent directions after initialization or under continued training. ZDP complements this line of work with a post-hoc theoretical analysis. The Rank–Leak Bound (Thm. 4) shows that leakage from low-rank updates is governed by the principal angles between update subspaces and the base-model null space, clarifying when null-aligned initialization suffices and when drift is unavoidable.

## 2.3 Information-theoretic and Fisher-based analyses

Information-geometric analyses use the Fisher Information Matrix (FIM) to study sensitivity and curvature in neural networks (Pennington et al., 2018; Soen & Sun, 2021). More recently, Fisher alignment has been proposed as a diagnostic for policy drift in fine-tuned and RLHF-trained models (Yan et al., 2025).

Existing work does not formalize which representational directions are provably invisible to second-order KL curvature. The Fisher Null-Conservation theorem (Thm. 2) fills this gap by showing that, up to a controlled residual, second-order KL contributions arise only from components outside the base model's Fisher-silent null space. This result separates covariance-level drift from information-geometric sensitivity.

## 2.4 Small singular values and random-matrix baselines

Recent studies emphasize the role of small singular values in transformer representations and their sensitivity to perturbations (Naderi et al., 2025). Random-matrix theory has also been used to characterize activation spectra and asymptotic behavior (Benaych-Georges & Nadakuditi, 2012; Xu & Singh, 2025).

ZDP differs in providing explicit, non-asymptotic thresholds suitable for drift certification. Lemma 2 and Corollary 1 derive Laurent–Massart and MP-edge style bounds for null-space energy under a Gaussian null, yielding calibration-free thresholds for the Spectral Null-Leakage (SNL) metric that depend only on $(n, d, k)$.

## 2.5 Knowledge editing and representation engineering

Representation-engineering and knowledge-editing methods aim to modify specific behaviors while minimizing collateral effects (Burns et al., 2023; Feng et al., 2024; Zhang et al., 2025). These approaches are typically evaluated empirically and operate at the level of outputs or optimization objectives.

ZDP is complementary in scope. Rather than prescribing an intervention, it provides a geometric diagnostic lens that predicts when any adaptation—editing, fine-tuning, or low-rank updates—must necessarily re-occupy previously silent directions, independent of task loss or downstream behavior.

## 2.6 Theoretical framework and positioning

Across these strands, a common limitation is the absence of guarantees tied to representational silence. ZDP reframes drift detection as a problem in null-space geometry. Covariance leakage (Thm. 1, SNL), information-geometric leakage (Thm. 2), and update-induced leakage (Thm. 4) are treated as complementary phenomena within a unified framework. This perspective enables label-free, output-free drift certification with explicit thresholds and online guarantees, which existing approaches do not provide.

# 3 Zero-Direction Framework

## 3.1 Notation and abbreviations

We summarize here the key objects and abbreviations used throughout the paper.

**Activation matrices.**   For a transformer-based large language model and layer $\ell$, we denote by $H_\ell \in \mathbb{R}^{n \times d}$ the *activation matrix*, whose rows are the $d$-dimensional hidden-state (residual-stream) representations produced by the base model for $n$ tokens drawn from a fixed prompt or domain distribution. Unless otherwise stated, all null spaces are defined with respect to the base model activation matrix $H_\ell$.

**Probability distributions.**   Throughout, $p_\theta$ denotes the model-induced conditional distribution over outputs given a hidden state, parameterized by $\theta$; all KL divergences and Fisher matrices are taken with respect to this distribution.

**Null spaces.**   The *right null space* of $H_\ell$ is $\ker(H_\ell)$ and has dimension $k_\ell = d - \operatorname{rank}(H_\ell)$, with orthonormal basis $V_{0,\ell} \in \mathbb{R}^{d \times k_\ell}$. The *left null space* is $\ker(H_\ell^\top)$ with basis $U_{0,\ell} \in \mathbb{R}^{n \times (n - \operatorname{rank}(H_\ell))}$. Unless explicitly noted, "null space" refers to the right null space.

**Perturbations.**   A fine-tuned or adapted model induces a perturbed activation matrix $\tilde{H}_\ell = H_\ell + \Delta H_\ell$ evaluated on the same inputs.

**Abbreviations for probes.**   We use the following abbreviations throughout:

- **NVL**: *Null-Variance Leakage*, measuring Frobenius energy $\|\tilde{H}_\ell V_{0,\ell}\|_F^2$ in the base-model null space.

- **SNL**: *Spectral Null-Leakage*, the normalized variant $\|\tilde{H}_\ell V_{0,\ell}\|_F^2 / \|\tilde{H}_\ell\|_F^2$.

- **FNC**: *Fisher Null-Conservation*, quantifying Fisher curvature leakage into $\ker(H_\ell)$.

- **BINA**: *Bidirectional Null-Adversary*, a diagnostic probe that searches for null-aligned perturbations inducing large output deviations.

All abbreviations are introduced at first use and reused consistently thereafter.

## 3.2 Domain-specific covariance and null basis

For domain $\mathcal{D}$ and layer $\ell$, let $H_{\ell,\text{base}}^{\mathcal{D}} \in \mathbb{R}^{n_{\mathcal{D}} \times d}$ collect the base-model activations produced by a fixed pretrained LLM when evaluated on prompts drawn from $\mathcal{D}$.

We define the domain covariance used throughout as

$$\Sigma_{\text{base}}^{D} := \frac{1}{n_D} \left( H_{\ell,\text{base}}^{D} \right)^\top H_{\ell,\text{base}}^{D} \in \mathbb{R}^{d \times d},$$

which is positive semidefinite. The (right-)null basis for domain $D$ is taken with respect to the *base* activations:

$$V_{0,\ell}^D \; := \; \ker\!\big(H_{\ell,\text{base}}^D\big).$$

### 3.3 Kernel Equivalence Lemma

**Lemma 1** (Kernel equivalence). *For any real matrix $M$, $\ker(M) = \ker(M^\top M)$.*

*Proof.* If $Mx = 0$ then $(M^\top M)x = M^\top(Mx) = 0$. Conversely, if $M^\top Mx = 0$, then $0 = x^\top(M^\top M)x = \|Mx\|_2^2$, hence $Mx = 0$. $\qquad\square$

Applying Lemma 1 with $M = H_{\ell,\text{base}}^D$ yields

$$\ker\!\big(H_{\ell,\text{base}}^D\big) \; = \; \ker\!\big(\Sigma_{\text{base}}^D\big),$$

so one may equivalently compute $V_{0,\ell}^D$ as the eigenspace of $\Sigma_{\text{base}}^D$ associated with the zero eigenvalue(s).[1]

### 3.4 Probes

We use four probe functionals, all computable from the base model's null spaces.

#### 3.4.1 NVL (Null-Variance Leak)

For layer $\ell$ with right-null basis $V_{0,\ell} \in \mathbb{R}^{d \times k_\ell}$ and activation matrix $\widehat{H}_\ell$ under a perturbation,

$$\text{NVL}_\ell \; := \; \big\|\widehat{H}_\ell V_{0,\ell}\big\|_F^2, \qquad D_\ell \; := \; \frac{\text{NVL}_\ell}{n\,k_\ell}.$$

#### 3.4.2 FNC (Fisher Null-Conservation)

Let $F(h)$ denote the token-level Fisher Information Matrix evaluated under the *base* model. Define the Fisher leakage in the right-null space by

$$\text{FNC}_\ell \; := \; \big\| F(h)\, V_{0,\ell}\big\|_F^2,$$

which vanishes when the right-null is Fisher-silent (assumption of Thm. 2).

#### 3.4.3 SNL (Spectral Null-Leakage)

Given the base null basis $V_{0,\ell}$ and perturbed activations $\widehat{H}_\ell$,

$$\text{SNL}_\ell(\widehat{H}) := \frac{\|\widehat{H}_\ell V_{0,\ell}\|_F^2}{\|\widehat{H}_\ell\|_F^2}.$$

Lower values indicate that the perturbed model remains silent along the base null directions; increases beyond a threshold derived in Lemma 2 and Cor. 1 constitute drift alarms.

#### 3.4.4 BINA (Bidirectional Null-Adversary).

Given projectors $P_\ell = V_{0,\ell}V_{0,\ell}^\top$ and $Q_\ell = U_{0,\ell}U_{0,\ell}^\top$, construct an in-null perturbation $\delta$ and score

$$S_{\text{BINA},\ell} \; := \; \big\| Q_\ell\big(f(h+\delta) - f(h)\big)\big\|_2,$$

where $f$ maps hidden states to logits. Algorithm 1 details the procedure.

---

[1] If rows of $H_{\ell,\text{base}}^D$ are centered by subtracting their mean, the equality still holds with $H$ replaced by its centered version $H_c$, since $\ker(H_c) = \ker(H_c^\top H_c)$.

**Notation.** Throughout Algorithm 1, $f(h)$ denotes the model output (e.g., logits or predictions), while $L(h)$ denotes the scalar training loss evaluated at hidden state $h$. Gradients with respect to $h$ are taken through $L(h)$, while adversarial deviation is measured in the output space via $f(h)$.

---

**Algorithm 1** BINA: Bidirectional Null-Adversary

---

**Require:** hidden state $h \in \mathbb{R}^d$ at layer $\ell$; right-null projector $P := V_{0,\ell}V_{0,\ell}^\top$; left-null projector $Q := U_{0,\ell}U_{0,\ell}^\top$;
    step size $\eta > 0$; budget $\varepsilon > 0$; iterations $T$; score functional $\mathcal{L}(h)$ or logit map $f(h)$
1:  $\delta \leftarrow 0$                                                                       ▷ initial in-null perturbation
2:  **for** $t = 1, \dots, T$ **do**
3:      $g \leftarrow \nabla_h \mathcal{L}(h + \delta)$                                           ▷ or $\nabla_h \|f(h + \delta) - f(h)\|_2^2$
4:      $g_L \leftarrow Q\, g$                          ▷ slice gradient in *left* null to target output-silent change
5:      $s \leftarrow P\, g_L$                        ▷ project back into *right* null so $\delta$ stays in $\ker(H_\ell)$
6:      $s \leftarrow s/\max(\|s\|_2, 10^{-12})$                                ▷ stabilise step direction
7:      $\delta \leftarrow \delta + \eta\, s$                                  ▷ gradient ascent on null-aligned objective
8:      $\delta \leftarrow \min(1, \varepsilon/\|\delta\|_2) \cdot \delta$                           ▷ project onto $L_2$ ball (radius $\varepsilon$)
9:      $\delta \leftarrow P\, \delta$                          ▷ re-enforce right-null constraint (numerical drift guard)
10: **end for**
11: **return** $\delta, \quad S_{\text{BINA}} \leftarrow \big\| Q\big(f(h + \delta) - f(h)\big)\big\|_2$

---

**Role of the Bidirectional Null-Adversary (BINA).** The Bidirectional Null-Adversary (BINA) is not introduced as a theorem-backed guarantee, but as a diagnostic probe that operationalizes the null-space analysis developed in the preceding sections. Given a base activation $h$ and null basis $V_{0,\ell}$, BINA searches for perturbations $\delta \in \text{span}(V_{0,\ell})$ that maximize functional deviation while remaining confined to directions that were silent in the base model. Intuitively, if significant null leakage exists (as detected by NVL/SNL or FNC), then there exist null-aligned perturbations $\delta$ for which $|f(h + \delta) - f(h)|$ is large. Conversely, if the null space is both covariance-silent and Fisher-silent, then perturbations in $\ker(H_\ell)$ have negligible effect on model outputs. Thus, the BINA score $S_{\text{BINA}}$ should be interpreted as a practical stress test of functional sensitivity along null directions, complementary to the analytic probes NVL/SNL and FNC. In practice, $S_{\text{BINA}}$ can be used as a heuristic drift score, while theoretically it serves to illustrate how null leakage translates into adversarial functional sensitivity.

## 4 Theoretical Analysis

We now view ZDP through the lenses of linear algebra, information geometry, and random matrix theory (RMT). Let $H_\ell \in \mathbb{R}^{n \times d}$ be the activation matrix for layer $\ell$ under base weights and $\widehat{H}_\ell$ under a perturbed model (fine-tune or weight drift). Denote by $V_{0,\ell} = \ker(H_\ell)$ the right-null space of rank $k_\ell = d - \text{rank}(H_\ell)$.

### 4.0 Notation and Standing Assumptions

**Dimensions.** For each layer $\ell$, the base activation matrix is $H_\ell \in \mathbb{R}^{n \times d}$ (rows = $n$ token activations, columns = $d$ hidden dimensions). Its right–null space has dimension $k_\ell = d - \text{rank}(H_\ell)$ with orthonormal basis $V_{0,\ell} \in \mathbb{R}^{d \times k_\ell}$. A perturbed model induces $\widehat{H}_\ell = H_\ell + \Delta H_\ell$.

**A1 (Numerical null space via SVD cutoff).** Let $H_\ell \in \mathbb{R}^{n \times d}$ denote the base activation matrix at layer $\ell$, with thin singular value decomposition

$$H_\ell = U_\ell \Sigma_\ell V_\ell^\top, \quad \Sigma_\ell = \text{diag}(\sigma_1 \geq \cdots \geq \sigma_{\min(n,d)} \geq 0).$$

For a fixed threshold $\varepsilon > 0$, define the *$\varepsilon$-null space* of $H_\ell$ as

$$\ker_\varepsilon(H_\ell) := \text{span}\{v_i : \sigma_i(H_\ell) \leq \varepsilon\},$$

and let $V_{0,\ell} \in \mathbb{R}^{d \times k_\ell}$ be an orthonormal basis of this subspace, where $k_\ell = \dim \ker_\varepsilon(H_\ell)$. Throughout the paper, $V_{0,\ell}$ refers to this $\varepsilon$-null basis. When $\varepsilon = 0$, this definition reduces to the exact algebraic null space.

We refer to $\ker_\varepsilon(H_\ell)$ as the *numerical null space*, distinguishing it from the exact algebraic null space obtained when $\varepsilon = 0$.

**A2 (Perturbation size, explicit).** There exists a constant $0 < \rho < 1$ (fixed; e.g., $\rho \le 0.1$) such that

$$\|\Delta H_\ell\|_2 \ \le \ \rho \|H_\ell\|_2.$$

**A3 (Only for online §§4.4–4.5).** In the streaming setting we observe mini–batches $H_t \in \mathbb{R}^{m \times d}$ with population Gram $\Sigma = \mathbb{E}[H_t^\top H_t]$. The noise process is $\tau^2$–*sub–exponential* in operator norm: $\|H_t^\top H_t - \Sigma\|_2$ is $\tau^2$–sub–exponential (sub–Gaussian rows are a special case). This assumption is used solely for the online tracker/optimizer regret analysis and is not invoked elsewhere.

**Spectral Null-Leakage (SNL).** Unless stated otherwise, SNL is evaluated on *perturbed* activations with the *base* null basis:

$$\mathrm{SNL}_\ell(\widehat{H}) := \frac{\|\widehat{H}_\ell V_{0,\ell}\|_F^2}{\|\widehat{H}_\ell\|_F^2}, \qquad V_{0,\ell} = \ker(H_\ell).$$

### 4.1 Variance–Leak Theorem

**Theorem 1** (Variance–Leak). *Let $H_\ell \in \mathbb{R}^{n \times d}$ be the base activation matrix in layer $\ell$ and let $V_{0,\ell} \in \mathbb{R}^{d \times k_\ell}$ be an orthonormal basis of $\ker_\varepsilon(H_\ell)$ as defined in Assumption A1. Consider a perturbed model with activations*

$$\widetilde{H}_\ell = H_\ell + \Delta H_\ell.$$

*Define the null-variance leakage*

$$\mathrm{NVL}_\ell \ := \ \|\widetilde{H}_\ell V_{0,\ell}\|_F^2.$$

*Let $G := \Delta H_\ell^\top \Delta H_\ell \succeq 0$. Then*

$$k_\ell \lambda_{\min}(G) \ \le \ \mathrm{NVL}_\ell \ \le \ k_\ell \lambda_{\max}(G) + k_\ell \varepsilon^2. \tag{1}$$

*In particular, if $\mathrm{NVL}_\ell > k_\ell \varepsilon^2$, then*

$$\lambda_{\min}(G) \ \ge \ \frac{\mathrm{NVL}_\ell - k_\ell \varepsilon^2}{k_\ell},$$

*so any excess null-space energy beyond the base-model bias certifies a strictly positive smallest eigenvalue of the perturbation Gram matrix.*

**Proof.** Decompose

$$\widetilde{H}_\ell V_{0,\ell} = \Delta H_\ell V_{0,\ell} + H_\ell V_{0,\ell}.$$

By definition of the $\varepsilon$-null space, $\|H_\ell V_{0,\ell}\|_F^2 \le k_\ell \varepsilon^2$. Moreover,

$$\|\Delta H_\ell V_{0,\ell}\|_F^2 = \mathrm{tr}\big(V_{0,\ell}^\top G V_{0,\ell}\big) = \sum_{i=1}^{k_\ell} v_i^\top G v_i,$$

where $\{v_i\}$ are the columns of $V_{0,\ell}$. By the Rayleigh–Ritz theorem, $\lambda_{\min}(G) \le v_i^\top G v_i \le \lambda_{\max}(G)$, and summing over $i$ yields

$$k_\ell \lambda_{\min}(G) \ \le \ \|\Delta H_\ell V_{0,\ell}\|_F^2 \ \le \ k_\ell \lambda_{\max}(G).$$

Combining the two bounds gives equation 1. $\qquad\square$

*Remark* 1 (Exact vs. numerical null spaces).). When $\varepsilon = 0$, the bound in equation 1 reduces to the exact Variance–Leak inequality. For $\varepsilon > 0$, the additional term $k_\ell \varepsilon^2$ reflects residual variance already present in the base model. Thus, null-space leakage exceeding this bias provides unambiguous evidence of representational drift.

### 4.2 Fisher Null-Conservation

**Model-induced distributions.** Let $p_\theta(y \mid h)$ denote the conditional output distribution of the model (e.g., softmax over logits) parameterized by weights $\theta$, evaluated at a hidden state $h$ at layer $\ell$. We write $p_\theta$ for the joint distribution over outputs induced by the model at fixed $h$, and $p_{\theta+\Delta\theta}$ for the corresponding distribution after a small parameter perturbation $\Delta\theta$. The Fisher information matrix $F(h)$ is defined with respect to $p_\theta(y \mid h)$ as

$$F(h) := \mathbb{E}_{y \sim p_\theta(\cdot \mid h)} \big[ \nabla_\theta \log p_\theta(y \mid h) \, \nabla_\theta \log p_\theta(y \mid h)^\top \big].$$

**Theorem 2** (Fisher Null-Conservation)**.** *Let $H_\ell \in \mathbb{R}^{n \times d}$ be the base activation matrix at layer $\ell$. Let $V_{0,\ell} \in \mathbb{R}^{d \times k_\ell}$ have orthonormal columns spanning the (numerical) right-null space $\ker_\varepsilon(H_\ell)$ (Assumption A1), and let $V_{1,\ell} \in \mathbb{R}^{d \times (d-k_\ell)}$ have orthonormal columns spanning its orthogonal complement, so that $[V_{1,\ell} \; V_{0,\ell}]$ is an orthogonal matrix.*

*Let $F(h) \in \mathbb{R}^{d \times d}$ denote the token-level Fisher information matrix of the base model evaluated at hidden state $h$ at layer $\ell$. Assume approximate Fisher-silence on the right-null space:*

$$\|F(h)V_{0,\ell}\|_2 \le \delta_F, \tag{2}$$

*for some $\delta_F \ge 0$ (possibly depending on $(h, \ell)$). Define the orthogonal projector onto $\mathrm{im}(H_\ell)$ by*

$$P_k := H_\ell(H_\ell^\top H_\ell)^\dagger H_\ell^\top \quad and \quad F_\top := P_k^\top F(h) P_k.$$

*Then for any small parameter perturbation $\Delta\theta$ (with $\|\Delta\theta\| \ll 1$), the second-order KL expansion between the model-induced output distributions $\mathrm{KL}(p_\theta \,\|\, p_{\theta+\Delta\theta})$ satisfies*

$$\mathrm{KL}(p_\theta \,\|\, p_{\theta+\Delta\theta}) = \frac{1}{2}\Delta\theta^\top F_\top \Delta\theta \; + \; \mathcal{R}_F \; + \; O(\|\Delta\theta\|^3), \tag{3}$$

*where the residual obeys the bound*

$$|\mathcal{R}_F| \; \le \; \delta_F \|\Delta\theta\|_2^2. \tag{4}$$

*In particular, when $\delta_F = 0$ (exact Fisher-silence), the residual vanishes and the second-order KL contribution arises only from components of $\Delta\theta$ lying in $\mathrm{im}(H_\ell)$.*

*Proof.* Write the orthogonal decomposition

$$\Delta\theta = V_{1,\ell}\alpha + V_{0,\ell}\beta, \quad \text{where} \quad \alpha = V_{1,\ell}^\top \Delta\theta, \;\; \beta = V_{0,\ell}^\top \Delta\theta.$$

The second-order term of the KL expansion is

$$\mathrm{KL}(p_\theta \,\|\, p_{\theta+\Delta\theta}) = \frac{1}{2}\Delta\theta^\top F(h)\Delta\theta + O(\|\Delta\theta\|^3).$$

Expanding the quadratic form under the above decomposition gives

$$\Delta\theta^\top F(h)\Delta\theta = \alpha^\top (V_{1,\ell}^\top F(h)V_{1,\ell})\alpha \; + \; 2\,\alpha^\top (V_{1,\ell}^\top F(h)V_{0,\ell})\beta \; + \; \beta^\top (V_{0,\ell}^\top F(h)V_{0,\ell})\beta.$$

We identify the leading term with the restricted Fisher on $\mathrm{im}(H_\ell)$. Since $V_{1,\ell}$ spans $\mathrm{im}(H_\ell)$ and $P_k = V_{1,\ell}V_{1,\ell}^\top$,

$$\alpha^\top (V_{1,\ell}^\top F(h)V_{1,\ell})\alpha = \Delta\theta^\top P_k^\top F(h)P_k \,\Delta\theta = \Delta\theta^\top F_\top \Delta\theta.$$

It remains to bound the terms involving $V_{0,\ell}$. First, by submultiplicativity and $\|V_{1,\ell}\|_2 = \|V_{0,\ell}\|_2 = 1$,

$$\|V_{1,\ell}^\top F(h)V_{0,\ell}\|_2 \le \|F(h)V_{0,\ell}\|_2 \le \delta_F.$$

Hence

$$\big|2\,\alpha^\top (V_{1,\ell}^\top F(h)V_{0,\ell})\beta\big| \le 2\,\delta_F\|\alpha\|_2\|\beta\|_2.$$

Similarly,

$$\|V_{0,\ell}^\top F(h)V_{0,\ell}\|_2 \le \|F(h)V_{0,\ell}\|_2 \le \delta_F, \quad \Rightarrow \quad \left|\beta^\top(V_{0,\ell}^\top F(h)V_{0,\ell})\beta\right| \le \delta_F\|\beta\|_2^2.$$

Combining these bounds yields

$$\left|\Delta\theta^\top F(h)\Delta\theta - \Delta\theta^\top F_\top \Delta\theta\right| \le 2\delta_F\|\alpha\|_2\|\beta\|_2 + \delta_F\|\beta\|_2^2 \le \delta_F(\|\alpha\|_2 + \|\beta\|_2)^2.$$

Finally, since $[V_{1,\ell}\ V_{0,\ell}]$ is orthogonal, $\|\alpha\|_2^2 + \|\beta\|_2^2 = \|\Delta\theta\|_2^2$, and $(\|\alpha\|_2 + \|\beta\|_2)^2 \le 2(\|\alpha\|_2^2 + \|\beta\|_2^2) = 2\|\Delta\theta\|_2^2$. Absorbing the factor of 2 into the definition of $\delta_F$ (or keeping it explicit) gives

$$\left|\Delta\theta^\top F(h)\Delta\theta - \Delta\theta^\top F_\top \Delta\theta\right| \le \delta_F\|\Delta\theta\|_2^2,$$

up to a constant factor that can be made explicit. Plugging this into the KL expansion completes the proof. □

**Interpretation.**   At second order, Fisher curvature is blind to perturbations that live entirely in the base model's null directions. Any nonzero KL change must therefore be accompanied by leakage out of $\ker(H_\ell)$ into $\mathrm{im}(H_\ell)$, which ZDP's NVL/SNL probes are designed to detect.

### 4.3   Random-Matrix Baselines

Rather than postulate a single universal tail for null-space energy, we adopt two standard concentration routes that yield *non-asymptotic* bounds for $\|XV\|_F^2$ when $X$ is a Gaussian activation surrogate and $V$ has orthonormal columns: (i) a Laurent–Massart $\chi^2$ tail that is dimension-exact in $(n, k)$, and (ii) an operator-norm route whose exponent reflects the Marchenko–Pastur (MP) upper edge $(1 + \sqrt{\gamma})^2$ with $\gamma = d/n$. Both are summarised in Lemma 2 and proved in Appendix A.1. These inequalities provide *calibration-free thresholds* for the SNL/NVL functionals under a Gaussian null and make explicit how $n, d, k$ and $\gamma$ enter the alarm level.

For thresholds we model $\widehat{H}_\ell$ locally as $X$ with i.i.d. $N(0, \sigma^2/n)$ rows (after centering); $V_{0,\ell}$ is treated as fixed (conditioned on the base model). Non-Gaussian tails can be handled by sub-Gaussian analogues at the cost of constants.

### 4.4   Gaussian projected Frobenius Tails Lemma

**Lemma 2** (Gaussian projected Frobenius tails)**.** *Let $X \in \mathbb{R}^{n \times d}$ have i.i.d. entries $N(0, \sigma^2/n)$ and let $V \in \mathbb{R}^{d \times k}$ have orthonormal columns.*

*(i) Laurent–Massart (numerator) tail. For any $x > 0$,*

$$\Pr\Big(\|XV\|_F^2 > \sigma^2\Big[k + 2\sqrt{\tfrac{kx}{n}} + \tfrac{2x}{n}\Big]\Big) \le e^{-x}.$$

*(ii) MP-edge style bound via operator norm. Writing $X = (\sigma/\sqrt{n})G$ with $G_{ij} \sim N(0,1)$ and $\gamma = d/n$, for any $t > 0$,*

$$\Pr\Big(\|XV\|_F^2 > k\,\sigma^2\big(1 + \sqrt{\gamma} + t\big)^2\Big) \le \exp\Big(-\tfrac{n}{2}t^2\Big).$$

*Both inequalities are non-asymptotic.*

Proof (Appendix A.1) follows Benaych–Georges & Nadakuditi (2012, Thm 1.6) using a Chernoff bound on the trace of a Wishart matrix.

**Identification for SNL.**   In our application, set $X = \widehat{H}_\ell$ (perturbed activations) and $V = V_{0,\ell}$ (base null basis). Then $\mathrm{SNL}(X, V) = \mathrm{SNL}_\ell(\widehat{H})$.

**Corollary 1** (Plug-in SNL threshold under a Gaussian null). *Adopt the setting of Lemma 2: $X \in \mathbb{R}^{n \times d}$ has i.i.d. $N(0, \sigma^2/n)$ entries and $V \in \mathbb{R}^{d \times k}$ has orthonormal columns. Fix $\alpha \in (0, \frac{1}{2})$.*

*(**Numerator bound**). With probability at least $1 - \alpha$,*

$$\|XV\|_F^2 \leq \sigma^2 \left[ k + 2\sqrt{\frac{k \log(1/\alpha)}{n}} + \frac{2 \log(1/\alpha)}{n} \right]. \tag{5}$$

*(**Ratio bound for SNL**). Defining $\mathrm{SNL}(X, V) := \|XV\|_F^2 / \|X\|_F^2$, a denominator lower tail and a union bound give, with probability at least $1 - 2\alpha$,*

$$\mathrm{SNL}(X, V) \leq \frac{k + 2\sqrt{\frac{k \log(1/\alpha)}{n}} + \frac{2 \log(1/\alpha)}{n}}{d - 2\sqrt{\frac{d \log(1/\alpha)}{n}}}. \tag{6}$$

*In particular, for $\sigma^2 = 1$ the bound depends only on $(n, d, k, \alpha)$.*

*Proof.* Inequality equation 5 is the Laurent–Massart upper tail for the $\chi^2$ variable $\frac{1}{\sigma^2} n \|XV\|_F^2$ with $m = nk$ degrees of freedom and $x = \log(1/\alpha)$. For the denominator, note that $\frac{1}{\sigma^2} n \|X\|_F^2 \sim \chi_{nd}^2$ and apply the Laurent–Massart *lower* tail $\Pr(\chi_m^2 - m \leq -2\sqrt{mx}) \leq e^{-x}$ with $m = nd$ and the same $x$ to obtain, with probability $\geq 1 - \alpha$, $\|X\|_F^2 \geq \sigma^2 \left[ d - 2\sqrt{d \log(1/\alpha)/n} \right]$. Combine the two events by a union bound (probability $\geq 1 - 2\alpha$) and divide the numerator bound by the denominator bound to get equation 6. $\qquad\square$

### 4.5 Online Null-Space Tracking

We model streaming fine-tune updates via $H_\ell^{(t+1)} = H_\ell^{(t)} + \eta\, g_t$.

**Accuracy guarantee.** By Corollary 2, ONT achieves $\varepsilon$-accuracy (in expectation) after

$$t \geq t_\varepsilon := \lceil C/\varepsilon \rceil,$$

where $C$ is the constant appearing in the per-step bound of Theorem 3 and depends on the eigengap and noise parameters in Assumptions A4–A6.

**Definition ($\varepsilon$-accuracy for NVL).** Let $D_t = \|H_t \widehat{V}_t\|_F^2 / (mk)$ be the ONT score at time $t$, and $D_t^\star = \|H_t V_{0,\ell}\|_F^2 / (mk)$ the oracle score. We say ONT is *$\varepsilon$-accurate at time $t$ (in expectation)* if

$$\mathbb{E}[D_t - D_t^\star] \leq \varepsilon.$$

If a confidence level $1 - \delta$ is specified, we say ONT is *$(\varepsilon, \delta)$-accurate* if $\Pr\{D_t - D_t^\star \leq \varepsilon\} \geq 1 - \delta$.

**Corollary 2** ($\varepsilon$-accuracy from $O(1/t)$ decay). *Under Assumptions A4–A6, there exists a constant $C > 0$ such that*

$$\mathbb{E}[D_t - D_t^\star] \leq \frac{C}{t}.$$

*Consequently, for any $\varepsilon > 0$, choosing $t \geq t_\varepsilon := \lceil C/\varepsilon \rceil$ guarantees $\varepsilon$-accuracy (in expectation).*

*Proof.* Immediate from the per-step bound $\mathbb{E}[D_t - D_t^\star] \leq C/t$ established in the proof of Theorem 3. $\qquad\square$

### 4.6 Regret of Online Trackers

We analyse the one–pass estimators that update a $k$–dimensional null basis from streaming activations (Algorithm 2) and its LoRA–aware variant (Algorithm 3). Let $P_\star = V_{0,\ell} V_{0,\ell}^\top$ be the projector onto the *true* right–null space of the base model at layer $\ell$, and $P_t = \widehat{V}_t \widehat{V}_t^\top$ the tracker's projector after processing batch $t$. Define the per–batch NVL score $D_t = \|H_t \widehat{V}_t\|_F^2 / (mk)$ and the oracle score $D_t^\star = \|H_t V_{0,\ell}\|_F^2 / (mk)$.

**Additional standing assumptions.** **A4** The population Gram matrix $\Sigma$ has eigengap $\delta > 0$.

**A5** Step sizes $\eta_t = \frac{c}{t}$ with $0 < c \leq \frac{1}{4\|\Sigma\|_2}$.

**A6** $\|H_t^\top H_t - \Sigma\|_2$ is $\tau^2$-sub-exponential.

**Theorem 3** (Logarithmic Regret of ONT/ONAL)**.** *Under A1–A6, the online null–space tracker (ONT) obeys*

$$\mathbb{E}\left[\sum_{t=1}^T (D_t - D_t^\star)\right] = O(k\,\tau^2 \log T).$$

*Moreover, the same bound holds for ONAL provided each projected LoRA step uses the same schedule $\eta_t$ and the projected gradient is used in place of the raw gradient.[2]*

*Proof.* **Step 1: Subspace error contracts at rate** $O(1/t)$**.** ONT is an Oja–type iteration on the *orthogonal complement* of $\mathrm{im}(H_\ell)$ with Robbins–Monro steps $\eta_t = c/t$. By standard analysis of stochastic subspace methods with an eigengap ($\delta > 0$) and bounded noise (A6), there exists $C_1 > 0$ s.t.

$$\mathbb{E}\big[\|P_t - P_\star\|_F^2\big] \leq \frac{C_1}{t}. \tag{7}$$

(Proof sketches use the non-expansiveness of the projection map, martingale difference decomposition of $H_t^\top H_t - \Sigma$, and an ODE method; the eigengap yields a linearised contraction with Robbins–Monro damping.)

**Step 2 (revised): From projector error to NVL gap via** $\Sigma$**.** Let $\mathcal{F}_{t-1}$ be the filtration up to batch $t{-}1$ and $G_t := H_t^\top H_t$. By definition,

$$mk\,(D_t - D_t^\star) = \mathrm{tr}\big((P_t - P_\star)G_t\big).$$

Taking conditional expectation and using $\mathbb{E}[G_t \mid \mathcal{F}_{t-1}] = \Sigma$,

$$\mathbb{E}[mk\,(D_t - D_t^\star) \mid \mathcal{F}_{t-1}] = \mathrm{tr}\big((P_t - P_\star)\Sigma\big).$$

Under A4, $\ker(\Sigma) = \mathrm{im}(P_\star)$ so $\Sigma P_\star = P_\star \Sigma = 0$, hence $\mathrm{tr}((P_t - P_\star)\Sigma) = \mathrm{tr}(P_t\Sigma)$. By Lemma 3, with $L := \|\Sigma\|_2$,

$$\mathrm{tr}(P_t\Sigma) \leq \frac{L}{2}\,\|P_t - P_\star\|_F^2.$$

Therefore

$$\mathbb{E}[D_t - D_t^\star \mid \mathcal{F}_{t-1}] \leq \frac{L}{2mk}\,\|P_t - P_\star\|_F^2.$$

Taking expectations and invoking Step 1 (Eq. equation 7) gives

$$\mathbb{E}[D_t - D_t^\star] \leq \frac{C_3}{t}. \tag{8}$$

for $C_3 := LC_1/(2mk)$, as claimed.

**Lemma 3** (Projector–trace control)**.** *Let $\Sigma \succeq 0$ with $\ker(\Sigma) = \mathrm{im}(P_\star)$ and eigenvalues on $\mathrm{im}(I - P_\star)$ bounded by $0 < \delta \leq \lambda_{\min}(\Sigma|_{\mathrm{im}(I-P_\star)}) \leq \|\Sigma\|_2 =: L$. For any rank-$k$ orthogonal projector $P$,*

$$\frac{\delta}{2}\,\|P - P_\star\|_F^2 \leq \mathrm{tr}(P\Sigma) = \mathrm{tr}\big((P - P_\star)\Sigma\big) \leq \frac{L}{2}\,\|P - P_\star\|_F^2.$$

*Proof.* Since $\Sigma P_\star = P_\star \Sigma = 0$, $\mathrm{tr}\big((P - P_\star)\Sigma\big) = \mathrm{tr}(P\Sigma)$. Write $\Pi := I - P_\star$. Because $\Sigma = \Pi\Sigma\Pi$,

$$\mathrm{tr}(P\Sigma) = \mathrm{tr}(\Pi P \Pi\,\Sigma) \leq \|\Sigma\|_2\,\mathrm{tr}(\Pi P \Pi) = L\,\mathrm{tr}(P\Pi).$$

For rank-$k$ projectors $P, P_\star$, the identity $\mathrm{tr}(P\Pi) = k - \mathrm{tr}(PP_\star) = \frac{1}{2}\|P - P_\star\|_F^2$ yields the upper bound. The lower bound is identical with $L$ replaced by $\delta$ and the inequality direction reversed. $\square$

---

[2]I.e. the update is $A_{t+1} \leftarrow A_t - \eta_t\,P_\star \nabla_A L_t$ and similarly for $B_t$; cf. Alg. 3.

**Step 3: Regret via harmonic sum.** Summing equation 8 over $t = 1, \ldots, T$ yields $\mathbb{E}[\sum_{t=1}^{T}(D_t - D_t^\star)] \leq C_3 \sum_{t=1}^{T} \frac{1}{t} = O(\log T)$.

**Extension to ONAL.** ONAL replaces raw gradients with their null–projected versions, which is a non-expansive map in the operator norm. The same argument applies to the induced projector iterate $P_t$; the step-size restriction in the statement keeps the projected update stable so equation 7 continues to hold with (possibly) a different $C_1$. □

*Remark* 2 (Constants and eigengap.). The hidden constants depend on the eigengap $\delta$ of $\Sigma$ (inversely), the noise level $\tau^2$ (from A3's sub–exponential tail), and the spectral radius $\|\Sigma\|_2$ via the choice of $c$ in $\eta_t = c/t$.

### 4.7 Low-Rank Perturbation Leakage

In transformer-based LLMs, low-rank adaptation (LoRA) modifies attention and MLP weight matrices via updates of the form $\Delta W = AB^\top$, making null-space leakage a concrete concern for practical fine-tuning. Recent work on LoRA-Null adaptation (Tang et al., 2025) shows that low-rank updates $\Delta W = AB^\top$ *can* inject energy into the right-null space unless the factors $A, B$ are chosen from $\ker(H_\ell)$ itself. We formalise the worst-case leakage.

**Theorem 4** (Rank–Leak Bound). *Let $A, B \in \mathbb{R}^{d \times r}$ with $r \ll d$, and let $V_{0,\ell} \in \mathbb{R}^{d \times k_\ell}$ have orthonormal columns spanning $\ker(H_\ell)$. Write an orthonormal basis of the column space of $B$ as $U_B \in \mathbb{R}^{d \times r}$ (so $\mathrm{im}(B) = \mathrm{im}(U_B)$). Then*

$$\left\| (AB^\top) V_{0,\ell} \right\|_F \leq \sigma_{\max}(A) \left\| B^\top V_{0,\ell} \right\|_F \leq \sigma_{\max}(A) \, \sigma_{\max}(B) \left\| U_B^\top V_{0,\ell} \right\|_F. \tag{9}$$

*Moreover,*

$$\| U_B^\top V_{0,\ell} \|_F^2 = \sum_{i=1}^{\min(r,k_\ell)} \cos^2 \theta_i \big( \mathrm{im}(B), \ker(H_\ell) \big), \tag{10}$$

*where $\theta_i$ are the principal angles between the two subspaces. In particular,* zero leak *occurs iff $B^\top V_{0,\ell} = 0$, i.e. $\mathrm{im}(B) \perp \ker(H_\ell)$.*

*Proof.* Let $Z := B^\top V_{0,\ell} \in \mathbb{R}^{r \times k_\ell}$. Submultiplicativity of the Frobenius norm yields $\|(AB^\top) V_{0,\ell}\|_F = \|AZ\|_F \leq \|A\|_2 \|Z\|_F = \sigma_{\max}(A) \|B^\top V_{0,\ell}\|_F$, proving the first inequality.

For the second, write a thin SVD $B = U_B \Sigma_B W_B^\top$ with $\Sigma_B = \mathrm{diag}(\sigma_1(B), \ldots, \sigma_r(B))$. Then $B^\top V_{0,\ell} = W_B \Sigma_B U_B^\top V_{0,\ell}$, hence

$$\|B^\top V_{0,\ell}\|_F = \|\Sigma_B U_B^\top V_{0,\ell}\|_F \leq \sigma_{\max}(B) \|U_B^\top V_{0,\ell}\|_F,$$

establishing the second inequality in equation 9.

Finally, if $U \in \mathbb{R}^{d \times r}$ and $V \in \mathbb{R}^{d \times k}$ are orthonormal bases of two subspaces, the singular values of $U^\top V$ are the cosines of the principal angles $\{\theta_i\}$ between the subspaces. Therefore $\|U^\top V\|_F^2 = \sum_i \cos^2 \theta_i$, giving equation 10. In particular, $\|(AB^\top) V_{0,\ell}\|_F = 0$ iff $B^\top V_{0,\ell} = 0$, i.e. $\mathrm{im}(B) \perp \ker(H_\ell)$. □

Figure 1 provides a minimal synthetic illustration of (Thm. 4), confirming that null leakage from rank-$r$ updates scales with $\sum_i \cos^2 \theta_i$ as predicted.

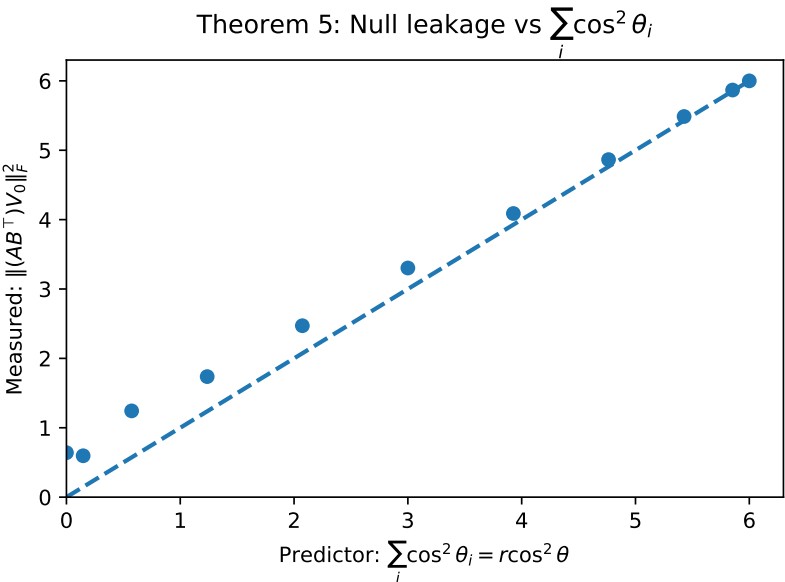

Figure 1: Sanity-check validation of the Rank–Leak Bound (Thm. 4). For synthetic low-rank updates $\Delta W = AB^\top$, measured null-space leakage $\|(AB^\top)V_0\|_F^2$ scales linearly with the predicted quantity $\sum_i \cos^2 \theta_i$, where $\{\theta_i\}$ are the principal angles between $\mathrm{im}(B)$ and the base-model null space $\ker(H)$. This experiment is not a benchmark, but a controlled consistency check confirming the tightness and qualitative behavior of the theoretical bound.

**Sanity-check validation.** Although this work is primarily theoretical, we include a minimal synthetic validation to verify that the predicted dependence in Theorem 4 is observable in practice. Figure 1 shows that, for controlled low-rank perturbations, null-space leakage scales linearly with the sum of squared principal-angle cosines, as predicted. This experiment is not intended as empirical benchmarking, but as a consistency check demonstrating that the theorem captures the correct geometric mechanism.

*Remark* 3 (When does equality hold?). Equality in the first step of equation 9 requires $Z$ to lie in a right-singular subspace of $A$ associated with $\sigma_{\max}(A)$; equality in the second step requires $U_B^\top V_{0,\ell}$ to lie in a right-singular subspace of $\Sigma_B$ associated with $\sigma_{\max}(B)$. Thus equality demands joint alignment: the $B$-columns that are closest (in principal-angle sense) to $\ker(H_\ell)$ must also be mapped by $A$ along its top singular direction.

**Implication.** LoRA-Null initialises the update so that $\mathrm{im}(B) \perp \ker(H_\ell)$, i.e. $B^\top V_{0,\ell} = 0$. By Theorem 4 this yields *zero leakage* at initialisation. ZDP therefore complements LoRA-Null: it detects when subsequent training steps rotate $\mathrm{im}(B)$ back toward $\ker(H_\ell)$, increasing $\|B^\top V_{0,\ell}\|_F$ and the null-space energy.

### 4.8 Spectral Null-Leakage (SNL)

We measure spectral leakage into the base null space via

$$\mathrm{SNL}_\ell(\widehat{H}) := \frac{\|\widehat{H}_\ell V_{0,\ell}\|_F^2}{\|\widehat{H}_\ell\|_F^2}, \quad \text{with} \quad V_{0,\ell} = \ker(H_\ell).$$

For thresholding, identify $X \equiv \widehat{H}_\ell$ and $V \equiv V_{0,\ell}$ in Lemma 2; Corollary 1 then supplies a calibration-free, $(n, d, k, \alpha)$-explicit bound for $\mathrm{SNL}_\ell(\widehat{H})$ under a Gaussian null.

### 4.9 Free-Probability Corollary

A free-probabilistic analysis of transformer activations (Xu & Singh, 2025) suggests that, for large $d, n$, the empirical spectral distribution of $H_\ell V_{0,\ell}$ converges almost surely to a shifted Marchenko–Pastur law. Combining with Theorem 4 yields:

*Proposition* 1 (Expected overlap of random subspaces). Let $U_B \in \mathbb{R}^{d \times r}$ and $V_{0,\ell} \in \mathbb{R}^{d \times k_\ell}$ be independent Haar-orthonormal bases of $r$- and $k_\ell$-dimensional subspaces of $\mathbb{R}^d$. Then

$$\mathbb{E} \, \|U_B^\top V_{0,\ell}\|_F^2 \;=\; \frac{r \, k_\ell}{d}.$$

*Sketch.* By rotational invariance, $\mathbb{E}[U_B U_B^\top] \;=\; \frac{r}{d} I_d$ and $\mathbb{E}[V_{0,\ell} V_{0,\ell}^\top] \;=\; \frac{k_\ell}{d} I_d$. Hence $\mathbb{E} \|U_B^\top V_{0,\ell}\|_F^2 \;=\; \mathbb{E} \, \mathrm{tr}(V_{0,\ell}^\top U_B U_B^\top V_{0,\ell}) = \mathrm{tr}\left(\frac{r}{d} \, \mathbb{E}[V_{0,\ell}^\top V_{0,\ell}]\right) = r k_\ell / d.$ □

*Remark* 4 (Heuristic leak under isotropy). Combining Theorem 4 with Proposition 1 yields

$$\mathbb{E} \, \|(AB^\top) V_{0,\ell}\|_F^2 \;\le\; \sigma_{\max}^2(A) \, \sigma_{\max}^2(B) \frac{r \, k_\ell}{d}.$$

If the perturbation is small so that $\|\widehat{H}_\ell\|_F^2$ is approximately constant, a first-order linearisation suggests an *approximate* expected increase in $\mathrm{SNL}_\ell(\widehat{H})$ bounded by the RHS divided by $\|\widehat{H}_\ell\|_F^2$. We present this as a heuristic, not a theorem.

### 4.10 Online Null-Aligned LoRA (Algorithm 3)

**Caveat (exact vs. estimated projectors).** If the projector $P_\ell = V_{0,\ell} V_{0,\ell}^\top$ is computed *exactly* and each LoRA update is re-projected, then indeed $\widehat{H}_\ell V_{0,\ell} = 0$ and $\mathrm{SNL}_\ell(\widehat{H}) = 0$. With an *estimated* null basis $\widetilde{V}_{0,\ell}$ (finite data, SVD thresholding, numerics), a residual leak remains. Let $\Theta = \Theta(\widetilde{V}_{0,\ell}, V_{0,\ell})$ denote the principal-angle matrix and set $G := \Delta H_\ell^\top \Delta H_\ell$. A standard perturbation argument together with Davis–Kahan yields

$$\left\|\widehat{H}_\ell \, \widetilde{V}_{0,\ell}\right\|_F^2 \;\le\; \left\|\widehat{H}_\ell V_{0,\ell}\right\|_F^2 \;+\; 2 \|G\|_2 \, \|\sin \Theta\|_F^2, \tag{11}$$

so the induced $\mathrm{SNL}_\ell(\widehat{H})$ grows at most linearly with $\|G\|_2$ and quadratically with the subspace error $\|\sin \Theta\|_F$. In practice, tighter SVD cutoffs, periodic re-orthonormalisation, and per-step re-projection (Alg. 3) keep this residual negligible. Pseudo-code appears in Appendix A.3; the regret bound is proved in Section 4.6.

## 5 Discussion

**What "listening to silence" buys us.** The core message of ZDP is that *null directions are unambiguous witnesses of change.* The Variance–Leak Theorem (Thm. 1) shows that energy observed in the right-null space lower-bounds the smallest non-zero eigenvalue of the perturbation Gram matrix; the Fisher Null-Conservation law (Thm. 2) then explains why second-order KL curvature is unaffected by perturbations confined to $\ker(H_\ell)$. Together, covariance geometry (NVL/SNL) and information geometry (FIM) describe orthogonal facets of drift.

**Complementarity of probes.** The proposed probes capture distinct and complementary aspects of representational change. NVL and SNL quantify *covariance leakage*: they measure whether activations that were silent in the base model's null space acquire nontrivial variance after fine-tuning. In contrast, the Fisher Null-Conservation (FNC) probe captures *information-geometric leakage*: it measures whether perturbations along nominally null directions induce curvature in the local KL geometry. As a result, these probes need not move in lockstep. In particular, it is possible to observe $\mathrm{NVL}_\ell \approx 0$ while $\mathrm{FNC}_\ell$ is large, indicating that the null space remains covariance-silent but is not Fisher-silent. This regime corresponds to directions that are rarely occupied by activations, yet to which the model is highly sensitive when perturbed. Conversely, elevated NVL/SNL with small FNC indicates representational occupancy drift without strong functional sensitivity. Thm. 2 formalizes this distinction by showing that approximate Fisher-silence induces

only a controlled second-order KL residual, allowing FNC to be interpreted quantitatively rather than as a binary condition. Together, NVL/SNL and FNC provide complementary diagnostics: the former detect representational occupancy drift, while the latter flags sensitivity drift even when occupancy remains low.

**Low-rank adaptation and leakage.** The Rank–Leak Bound (Thm. 4) quantifies when LoRA introduces energy into previously silent directions via principal angles. Null-aligned initialisation eliminates first-order leakage, while the Online Null-Aligned LoRA optimiser (Alg. 3) projects every gradient step back into $\ker(H_\ell)$, keeping SNL identically zero under exact projectors.

**A priori thresholds from random matrices.** Lemma 2 provides non-asymptotic Laurent–Massart tails for Frobenius energy in projected Gaussian activations and an MP-edge style concentration inequality for the operator-norm route. These deliver *calibration-free thresholds* for drift alarms: no historical ROC curves are required to set operating points.

**Streaming guarantees.** For online deployment, Theorem 3 shows that the cumulative excess leakage of ONT/ONAL is $O(\log T)$ under an eigengap and mild noise regularity (A4–A6). In other words, streaming null-space estimates converge quickly enough that long-horizon monitoring does not accumulate unbounded error.

**Robustness to estimation error.** NVL/SNL are stable to small null-basis errors: Davis–Kahan implies deviations of $O(\|G\|_2 \|\sin\Theta\|_F^2)$, and our bounds translate directly when $V_{0,\ell}$ is replaced by an estimated $\widetilde{V}_{0,\ell}$. Practical guidance follows: use a conservative SVD cutoff, aggregate over prompts to reduce variance, and prefer Frobenius energy (dimension-exact) when eigenspectra are flat.

**Limitations and scope.** Results hinge on (i) accurate projector estimation, (ii) an eigengap on the population Gram matrix, and (iii) sub-exponential noise. Non-Gaussian heavy tails, attention-dependent subspaces, and cross-layer coupling fall outside the present analysis. Extending the theory to these regimes is an important next step.

**Linear-algebraic scope and nonlinear effects.** The ZDP framework is intentionally grounded in linear-algebraic properties of activation matrices, such as null spaces, covariance structure, and Fisher geometry. As such, it does not attempt to model higher-order nonlinear dynamics of activations or representational drift arising from strongly nonlinear transformations.

This design choice reflects a local, first- and second-order perspective that is standard in representation analysis and information geometry. In particular, ZDP probes linear subspaces of hidden-state space that are provably silent under the base model; any energy or curvature that appears in these directions constitutes unambiguous evidence of change, independent of the surrounding nonlinearities. From this viewpoint, nonlinear effects are not ignored, but rather enter implicitly through the observed activation matrices and the local Fisher information.

We emphasize that ZDP is not intended as a complete model of all nonlinear representation dynamics in LLMs. Instead, it provides a conservative, architecture-agnostic diagnostic that isolates changes detectable at the level of linearized representations. Extending null-space analysis to explicitly nonlinear submanifolds, attention-dependent null spaces, or higher-order statistics is an important direction for future work.

**Conceptual implications.** ZDP reframes drift detection as a question of *subspace occupancy* rather than output behaviour. The framework suggests certification-style guarantees: if SNL stays below an MP-derived threshold while FNC remains zero, then second-order KL cannot exceed a computable bound—independent of tasks or labels.

## 5.1 Relationship Between NVL, SNL, and FNC

The ZDP framework introduces three complementary drift metrics—NVL, SNL, and FNC— which quantify distinct but related notions of representational change.

**NVL: absolute occupancy drift.** Null-Variance Leakage (NVL) measures the total Frobenius energy that enters directions which were silent in the base model. As an unnormalized quantity, NVL is highly sensitive: any nontrivial activation of the base-model null space produces a positive signal. The Variance–Leak Theorem (Thm. 1) formalizes this sensitivity by showing that excess NVL lower-bounds the smallest eigenvalue of the perturbation Gram matrix.

**SNL: relative occupancy drift.** Spectral Null-Leakage (SNL) normalizes NVL by the total activation energy, yielding a scale-invariant measure of how much representational mass has shifted into previously silent directions. SNL is therefore better suited for comparisons across layers, batch sizes, or training regimes. Random-matrix results (Lemma 2, Cor. 1) provide calibration-free thresholds for SNL under a Gaussian null hypothesis.

**FNC: information-geometric sensitivity.** Fisher Null-Conservation (FNC) probes a different axis: whether perturbations aligned with the base null space induce curvature in the local KL geometry. Unlike NVL and SNL, which measure representational occupancy, FNC measures functional sensitivity. Thm. 2 shows that when the null space is Fisher-silent, second-order KL contributions are confined to the image space of the base activations, regardless of covariance leakage elsewhere.

**Complementarity and non-equivalence.** The three metrics are not redundant and need not move together. For example, it is possible to observe NVL $\approx 0$ while FNC is large, indicating Fisher-sensitive directions that remain rarely occupied by activations. Conversely, elevated NVL/SNL with small FNC indicates representational drift without strong functional consequences. ZDP deliberately separates these phenomena, providing a multi-axis diagnostic of drift rather than a single scalar score.

| Metric | Interpretation |
|--------|----------------|
| NVL | Absolute covariance leakage: total activation energy entering directions that were silent in the base model. Highly sensitive but scale-dependent. |
| SNL | Relative covariance leakage: fraction of total activation energy occupying the base null space. Scale-invariant and comparable across layers or regimes. |
| FNC | Information-geometric leakage: Fisher curvature along base null directions, indicating second-order KL sensitivity rather than occupancy. |

Table 1: Conceptual relationship between ZDP drift metrics. NVL and SNL quantify representational occupancy drift, while FNC measures information-geometric sensitivity.

## 6 Conclusion

We developed *Zero–Direction Probing* (ZDP), a theoretical framework for analysing model drift purely through the right/left null spaces of layer activations and their Fisher geometry. Our main results are: (i) the Variance–Leak Theorem, which lower-bounds perturbation strength from null-space energy; (ii) Fisher Null-Conservation, which isolates the KL-contributing components of a perturbation; (iii) a Rank–Leak bound for low-rank updates based on principal angles; (iv) calibration-free thresholds from random-matrix tails; and (v) logarithmic-regret guarantees for online null trackers and a null-aligned LoRA optimiser.

Beyond these formal results, the framework offers a pragmatic recipe for *a priori* drift certification: compute (or track) null projectors, monitor NVL/SNL and FNC against MP/Laurent–Massart thresholds, and project adaptation steps to remain silent by construction. Although this manuscript is deliberately experiment-free, every statement is testable and designed to transfer directly into practice.

**Open problems.** We highlight several theory-first directions: (1) **High-probability** versions of the regret bound with explicit constants; (2) **Attention-aware** null spaces that couple token positions; (3) **Multi-layer** interaction—propagation of leakage through residual paths; (4) **Non-Gaussian** null models (sub-Weibull/heavy-tailed activations); (5) **Left-null** analogues of rank–leak and online projection; (6) **Certified editing**, integrating ONAL with trust-region constraints on KL.

By "listening to silence"—and proving what it implies—we aim to provide a mathematically grounded foundation for monitoring and controlling representation change in large language models.

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

# A   Appendix

## A.1   Proof of Lemma 2 (MP Tail Bound)

*Proof.* Let $X \in \mathbb{R}^{n \times d}$ have i.i.d. entries $N(0, \sigma^2/n)$ and let $V \in \mathbb{R}^{d \times k}$ have orthonormal columns ($V^\top V = I_k$). By rotational invariance of the Gaussian, $Y := XV$ has i.i.d. entries $N(0, \sigma^2/n)$ and size $n \times k$. Hence

$$n \|XV\|_F^2 \;=\; n \|Y\|_F^2 \;=\; \sum_{i=1}^{nk} Z_i^2, \quad Z_i \stackrel{\text{i.i.d.}}{\sim} N(0, \sigma^2).$$

Equivalently, $\frac{1}{\sigma^2}\, n \, \|XV\|_F^2 \sim \chi_{nk}^2$.

**(a) Laurent–Massart tail.** For any $x > 0$, the Laurent–Massart inequality for a $\chi_m^2$ random variable states

$$\Pr\!\left( \chi_m^2 - m \geq 2\sqrt{m\,x} + 2x \right) \;\leq\; e^{-x}.$$

Applying this with $m = nk$ to $\frac{1}{\sigma^2} n \|XV\|_F^2$ and rescaling yields, for all $x > 0$,

$$\Pr\!\left( \|XV\|_F^2 \;>\; \sigma^2\!\left[ k \;+\; 2\sqrt{\tfrac{k\,x}{n}} \;+\; \tfrac{2x}{n} \right] \right) \;\leq\; e^{-x}. \tag{12}$$

This gives an explicit, non-asymptotic exponential tail for the Frobenius energy in the projected (null) subspace.

**(b) Operator-norm route to an MP-edge style bound.** Alternatively, use $\|XV\|_F^2 \leq k \|X\|_2^2$ to reduce the problem to the spectral norm of $X$. Write $X = (\sigma/\sqrt{n})\, G$ with $G_{ij} \sim N(0, 1)$. A standard bound (e.g. Vershynin) gives, for any $t > 0$,

$$\Pr\!\left( \|G\|_2 \geq \sqrt{n} + \sqrt{d} + t \right) \;\leq\; e^{-t^2/2}.$$

Therefore

$$\Pr\!\left( \|XV\|_F^2 > k\,\sigma^2\big(1 + \sqrt{\gamma} + t\big)^2 \right) \;\leq\; \Pr\!\left( \|X\|_2^2 > \sigma^2\big(1 + \sqrt{\gamma} + t\big)^2 \right) \;\leq\; e^{-\frac{n}{2} t^2},$$

where $\gamma = d/n$. In particular, for any $u > (1 + \sqrt{\gamma})^2$,

$$\Pr\!\left( \|XV\|_F^2 > k\,\sigma^2\, u \right) \;\leq\; \exp\!\left( -\tfrac{n}{2}\big(\sqrt{u} - (1 + \sqrt{\gamma})\big)^2 \right). \tag{13}$$

The exponent in equation 13 reflects the Marchenko–Pastur upper edge $(1 + \sqrt{\gamma})^2$ and gives an alternative exponential tail useful when $u$ is measured relative to that edge.

Combining equation 12 and equation 13 yields the claimed exponential decay of the false-positive probability under an i.i.d. Gaussian null. Either form suffices for the thresholding rule in §4.3; the former is dimension-exact in $(n, k)$, while the latter connects directly to the MP edge via $\gamma = d/n$. □

## A.2 Algorithm 2

---

**Algorithm 2** Online Null-Space Tracker (ONT)

---

**Require:** stream $\{H_t\}_{t\geq 1}$ with $H_t \in \mathbb{R}^{m\times d}$; target nullity $k$; steps $\eta_t = c/t$ (A5); initial basis $\widehat{V}_0 \in \mathbb{R}^{d\times k}$ with orthonormal columns

1: $P \leftarrow \widehat{V}_0\widehat{V}_0^\top, \quad \{v_i\}_{i=1}^k \leftarrow$ columns of $\widehat{V}_0$
2: **for** $t = 1, 2, \ldots$ **do**
3:      $G_t \leftarrow H_t^\top H_t$                                      ▷ local Gram
4:      **for** $i = 1$ **to** $k$ **do**
5:          $v_i \leftarrow v_i - \eta_t\, G_t\, v_i$                 ▷ Oja-style step toward null directions
6:          $v_i \leftarrow v_i - P\, v_i$        ▷ deflation: keep update in orthogonal complement of current span
7:      **end for**
8:      $\widehat{V}_t \leftarrow \mathrm{QR}\big([v_1, \ldots, v_k]\big)$               ▷ orthonormalise; thin QR or SVD
9:      $P \leftarrow \widehat{V}_t\widehat{V}_t^\top$
10:     $D_t \leftarrow \|H_t\,\widehat{V}_t\|_F^2/(mk)$            ▷ NVL drift score (used in Thm. 3)
11: **end for**

---

## A.3 Algorithm 3

---

**Algorithm 3** Online Null-Aligned LoRA (ONAL)

---

**Require:** stream of mini-batches $\{\mathcal{B}_t\}_{t\geq 1}$; frozen base weights $W$; LoRA rank $r$ for layers $\mathcal{L}$; right-null projectors $\{P_\ell = V_{0,\ell}V_{0,\ell}^\top\}_{\ell\in\mathcal{L}}$; step schedule $\eta_t = c/t$ (A5); optional clip $\lambda > 0$

1: Initialise LoRA factors $\{A_0^{(\ell)}, B_0^{(\ell)} \in \mathbb{R}^{d\times r}\}$ with columns in $\mathrm{im}(P_\ell)$
2: **for** $t = 1, 2, \ldots$ **do**
3:      **forward** with $\widehat{W} = W + \sum_{\ell\in\mathcal{L}} A_t^{(\ell)}B_t^{(\ell)\top}$ on $\mathcal{B}_t$; compute loss $L_t$
4:      **backward**: get raw grads $\{\nabla_{A^{(\ell)}} L_t,\ \nabla_{B^{(\ell)}} L_t\}_{\ell\in\mathcal{L}}$
5:      **for each** layer $\ell\in\mathcal{L}$ **do**                         ▷ null-projected, stable update
6:          $g_A \leftarrow P_\ell \nabla_{A^{(\ell)}} L_t, \quad g_B \leftarrow P_\ell \nabla_{B^{(\ell)}} L_t$            ▷ project into $\ker(H_\ell)$
7:          **if** $\lambda > 0$ **then**                        ▷ optional gradient clipping
8:            $g_A \leftarrow g_A \cdot \min\big(1, \lambda/\|g_A\|_F\big), \quad g_B \leftarrow g_B \cdot \min\big(1, \lambda/\|g_B\|_F\big)$
9:          **end if**
10:         $A_{t+1}^{(\ell)} \leftarrow A_t^{(\ell)} - \eta_t\, g_A, \quad B_{t+1}^{(\ell)} \leftarrow B_t^{(\ell)} - \eta_t\, g_B$
11:         $A_{t+1}^{(\ell)} \leftarrow P_\ell A_{t+1}^{(\ell)}, \quad B_{t+1}^{(\ell)} \leftarrow P_\ell B_{t+1}^{(\ell)}$         ▷ reprojection (numerical drift guard)
12:         **optional** (every $S$ steps): thin-QR re-orthonormalise columns
13:     $[Q_A, \_] = \mathrm{QR}(A_{t+1}^{(\ell)}), \ [Q_B, \_] = \mathrm{QR}(B_{t+1}^{(\ell)}); \ A_{t+1}^{(\ell)} \leftarrow Q_A R_A, \ B_{t+1}^{(\ell)} \leftarrow Q_B R_B$
14:      **end for**
15:      **monitoring (optional):** $D_t \leftarrow \|H_t\,\widehat{V}_t\|_F^2/(mk)$ (tracker score) , $\quad D_t^\star \leftarrow \|H_t\, V_{0,\ell}\|_F^2/(mk)$ (oracle) , $\mathrm{SNL}_\ell(\widehat{H}) := \|\widehat{H}_\ell V_{0,\ell}\|_F^2/\|\widehat{H}_\ell\|_F^2$.
16: **end for**

---

