# OpenReview forum: "Zero-Direction Probing: A Linear-Algebraic Framework for Deep Analysis of Large- Language-Model Drift"
_TMLR — Rejected by TMLR_

### Review · Reviewer_8kSG · 2025-09-07

**Summary Of Contributions:**

This paper proposes Zero-Direction Probing (ZDP), a theoretical framework to quantify representational drift in Large Language Models (LLMs) by examining the null spaces of a base model’s layer activations. The work is theory-only and introduces several linear-algebraic metrics and guarantees for detecting when a fine-tuned (or otherwise perturbed) model has deviated from its original representations, without requiring any task labels or output-based evaluation. Key contributions/strengths:

- **Null-space Definitions & Metrics**: The authors formalize the use of right-null (input-silent) and left-null (output-silent) subspaces of transformer layer activations. They define Null-Variance Leak (NVL) as the Frobenius energy that a perturbed model’s activations inject into the base model’s right-null space, and a normalized variant Spectral Null-Leakage (SNL) which is the fraction of total activation energy lying in the base null subspace. They also define Fisher Null-Conservation (FNC) which measures how much of the base model’s Fisher information “leaks” into null directions, and propose BINA (Bidirectional Null-Adversary), an algorithm that actively probes the model by injecting perturbations in the base null space to amplify any output differences.
- **Theoretical Drift Bounds** : Under clearly stated assumptions (A1–A6) about the base model’s activation rank, perturbation size, eigengaps, and noise regularity, the paper proves a series of theorems quantifying drift:
1. Variance–Leak Theorem (NVL)
2. Fisher Null-Conservation (FNC)
3. Rank–Leak Bound
4. Spectral Null-Leakage & Random Matrix Thresholds
5. Online Tracking & Log-Regret
- **Conceptual Insights**: The theory yields several high-level takeaways. Notably, it separates covariance geometry vs. information geometry: NVL and SNL measure covariance change (new variance in silent directions), whereas the Fisher result connects to information-theoretic change. The Rank–Leak analysis explains in principled terms when parameter-efficient fine-tuning (like LoRA) will inadvertently utilize previously unused directions. The random matrix baseline provides a null-hypothesis for drift detection without requiring heuristic calibration, which is important for practical deployment of these probes. Overall, the work suggests that “listening to silence”, that is monitoring the activations that were zero-variance in the base model, can reveal representational changes with provable guarantees.

-----

Key weaknesses:

- **No empirical validation**:  The paper is entirely theory-only. While the proofs are rigorous, there is no empirical or even synthetic demonstration (e.g., NVL/SNL behavior on toy activations). This makes it harder to judge the practical impact of the results.

- **Ambiguity in defining null spaces**: Assumption A1 requires exact zero-variance directions $V_{0,\ell} = \ker(H_\ell), \quad H_\ell \in \mathbb{R}^{n \times d}$. In practice, models usually have very small (but nonzero) singular values rather than exact zeros. The paper does not explain how to choose a cutoff for approximate null spaces, leaving application unclear.

- **Strong Fisher-silence assumption**: Theorem 3 assumes $F(h) V_{0,\ell} = 0,$ i.e., the Fisher Information Matrix is strictly silent on the null space. In realistic settings, $\|F(h)V_{0,\ell}\|_F > 0$ may hold. The manuscript does not discuss how conclusions change if this assumption is only approximately true.

- **Limited discussion of BINA**: The Bidirectional Null-Adversary (BINA) is presented in Algorithm 1 but not theoretically analyzed. Its intended purpose (practical probe vs. adversarial stress test) is not made clear, unlike the other metrics with formal guarantees.

- **Presentation issues (minor)**: Assumptions A1–A6 are scattered across sections rather than summarized in one place. Theorem numbering skips (no Theorem 2). Some definitions (e.g., SNL) are repeated instead of referenced, which can confuse readers.

**Audience:**

Yes

**Audience Explanation:**

- **Relevance to LLM safety and fine-tuning**: Provides tools to detect representational drift during adaptation, relevant for preventing catastrophic forgetting and ensuring robustness without labeled data.
- **Theoretical novelty**: Shifts focus from high-variance subspaces (e.g., SVCCA, CKA) to null spaces, introducing rigorous bounds. This fresh angle will interest theorists in representation geometry and high-dimensional statistics.
- **Connections to practice**: Directly relates to parameter-efficient fine-tuning and editing methods (LoRA, LoRA-Null, AlphaEdit). The Rank–Leak bound offers theoretical foundations for when such methods succeed or fail.
- **General ML audience**: Useful for interpretability, diagnostics, and continual learning. NVL/SNL metrics and online tracking guarantees could be applied broadly since they require no labels.

**Overall**: The topic intersects theory, practice, and safety, making it highly relevant for TMLR readers across multiple subfields.

**Broader Impact Concerns:**

The paper’s contributions are primarily theoretical and aimed at diagnosing model changes, which generally carries positive broader impacts. Shortly:

- **Positive**: Enables safer deployment, interpretability, and monitoring of LLM fine-tuning without labels. It could help detect catastrophic forgetting or unintended drift.

- **Risks** (limited): Adversaries might evade detection by confining updates to base subspace, but this is difficult in practice. ZDP should be combined with other methods for complete coverage.

**Claims And Evidence:**

Yes

**Claims Explanation:**

- **Correctness of theoretical claims**: Theorems 1, 3, 4, and 5 (plus lemmas) are rigorously derived using standard tools (Rayleigh–Ritz, block decomposition of the FIM, singular value arguments, stochastic approximation). Proofs are step-by-step, with only minor steps omitted that are standard in the literature. I personally found no logical gaps or mathematical errors.
- **Clarity of assumptions**: Assumptions A1–A6 are clearly stated and generally realistic for theory work: A1 (null space well-defined), A2 (bounded perturbation), A3–A6 (eigengap and concentration for online guarantees).
- **Evidence provided**: As a theory paper, evidence consists of proofs and references to established results.
- **Limitation**: No empirical validation is provided. Practical usefulness is deferred to companion work. While this is acceptable given the theoretical focus, it leaves open how well the results translate to real models.


**Overall**: The claims are well-supported by rigorous proofs under explicit assumptions, with clear alignment between what is stated and what is proven. The absence of experiments does not undermine the internal validity, though it limits immediate practical confirmation.

**Requested Changes:**

I present my recommended changes:

**Critical**

1. **Clarify applicability of null-space assumptions.** Assumption A1 presumes the base model has exact zero-variance directions: $V_{0,\ell} = \ker(H_\ell), \quad H_\ell \in \mathbb{R}^{n \times d}.$ In practice, models usually have very small but non-zero singular values rather than exact zeros. Authors should explain how to choose an SVD cutoff ε (e.g., treating eigenvalues below a fraction of the max as null), and discuss the sensitivity of results if these directions have small variance. Also, please clarify how this impacts Theorem 1 (Variance–Leak) and the SNL thresholds from Lemma 2.

2. **Add an empirical illustration.** While the work is theory-only, a toy experiment would strengthen intuition. For example, fine-tune a small synthetic matrix $H$ with a rank-r update and show how NVL/SNL detect drift. Also, you can compare SNL values to thresholds from Corollary 1 under a Gaussian null. Furthermore, demonstrate Theorem 5’s prediction: measured null leakage matches $cos^2(\theta)$ principal angles. Even a single figure would validate that the metrics behave as expected and make the theory more accessible.

**Not so critical**

1. **Discuss the Fisher-silence assumption.** Theorem 3 assumes $F(h) V_{0,\ell} = 0$. In real models, $|F(h)V_{0,\ell}|$ may be small but not zero. Clarify whether $FNC_{\ell}$ can be measured empirically and how to interpret cases where NVL $\approx 0$ but FNC is large. This would guide practitioners in using FNC as a diagnostic tool.

2. **Expand on BINA's role.** The Bidirectional Null-Adversary (BINA) is introduced but not tied to a theorem. Clarify whether $S_{BINA}$ is is intended as a practical drift score or a conceptual probe. A short explanation such as: "If significant null leakage exists, BINA finds $\delta $ with
$ \|f(h+\delta) - f(h)\| $ large; otherwise, perturbations in $ \ker(H) $ have negligible effect" would help. Also, please clarify the notation in Algorithm 1 the difference between $f(h)$ and $L(h)$.

3. **Notation improvements.**
- Consolidate assumptions A1–A6 in one place (currently scattered).
- Fix numbering (there is no “Theorem 2”).
- Avoid redefining SNL; instead, reference its earlier definition.
- Clarify Algorithm 1, Step 8.

---

### Review · Reviewer_LZ2c · 2025-09-15

**Summary Of Contributions:**

This paper characterizes model drift by examining null directions in transformer activations, eliminating the need for task labels or output evaluation. By the right/left null spaces and their Fisher geometry, it's derived with derive concrete, testable guarantees on how representations evolve.

Strengthens:
1. Introduce a theoretical framework with information geometry to understand model drift.
2. The authors carried out a rigorous proof for their statements.

Weakness:
1. Related works and theoretical framework are more like a list of claims, not a part of well-structured paper. It makes a difficult to connect the theoretical results in this paper to a broader research context.
2. The definition and abbreviation are not clearly stated, which makes it difficult for readers to follow the results of the paper.
3. The connection between this framework and LLM is not clear.
4. Though authors claim that "empirical validation and benchmarking are deferred to companion work", it's necessary to validate the proposed algorithm in the paper.

**Audience:**

Yes

**Audience Explanation:**

This paper provides theoretical results with practical algorithm design for LLM drift, which may be of interest to our community.

**Broader Impact Concerns:**

No.

**Claims And Evidence:**

Yes

**Claims Explanation:**

The claims are supported by proofs.

**Requested Changes:**

Included in Summary of Contributions.

---

### Review · Reviewer_jQso · 2025-11-28

**Summary Of Contributions:**

This work introduces a theoretical framework called **Zero-Direction Probing (ZDP)** for analyzing drift in large language models after parameter updates (e.g., fine-tuning). The core idea is to study how model updates affect the **null directions** of transformer activations or in another word, those directions that previously had zero variance. The key advantage of this framework is that it analyzes how the null space changes **without requiring access to labels or outputs**. The authors define several probing functionals, including **NVL** (Null-Variance Leak), **FNC** (Fisher Null-Conservation), **SNL** (Spectral Null-Leakage), and **BINA** (Bidirectional Null-Adversary). The paper also provides multiple theoretical results:

1. **Variance Leak Theorem**: establishes bounds showing that when the model is perturbed, previously silent (null-space) directions can become activated.

2. **Fisher Null-Conservation Theorem**: shows that perturbations restricted to the base model’s null space do not induce second-order changes in KL divergence.

3. **Logarithmic Regret of ONT/ONAL**: proves that the cumulative excess null-space leakage over time scales linearly with the null-space dimension but only logarithmically with time.

4. **Rank-Leak Bound Theorem**: This theorem provides a mathematical limit on how much low-rank updates, such as those in LoRA, can leak energy into the null space of a model's activations,

**Audience:**

Yes

**Audience Explanation:**

The paper’s findings on detecting model drift through null-space leakage provide new theoretical tools that are directly relevant to the ongoing recent researches on interpretability, fine-tuning dynamics, and continual learning.

**Claims And Evidence:**

Yes

**Claims Explanation:**

The theoretical claim of the paper are proven and justified. The experimental aspect of the claims are omitted since the focus of work is on theory.

**Requested Changes:**

**Strengths:**

- The main idea of monitoring the null space to detect drift is interesting and could help us better understand how updates affect transformers during fine-tuning.
- The authors provide rigorous mathematical results, and the claims are well supported by theory.
- The metrics introduced in the paper could be useful in practice if we can find reliable ways to monitor them during real-world LLM training.

**Weaknesses:**

- The paper lacks explanation and intuition for many of the metrics (especially Section 3.3) and the theorems. Adding more discussion and intuitive examples would make the contributions much clearer.
- Some assumptions are very strong and may not hold in practice. For example, estimating the null space might be infeasible or too expensive, and it is not clear how errors in estimating the null space would affect the drift measurements. The Gaussian null model may also be unrealistic in practice, though this is less of a concern given the theoretical focus of the paper.
- The paper does not clearly define \(p_\theta\) and \(p_{\hat\theta}\) in Theorem 3 (or I did not find it at least). A short description or explicit definition would help the reader.
- It is unclear how the different metrics relate to each other intuitively. For example, how NVL, SNL, and FNC compare in measuring drift, or which metric is more sensitive in practice.
- The framework mainly focuses on linear-algebraic properties; non-linear aspects of activations or drift are not captured, which might limit the applicability to real LLM behavior.

---

### Author Response · Authors · 2025-12-14
**Summary of Revisions in Response to Reviewer Feedback**

Dear Reviewers,

Thank you for your careful reading of our manuscript and for the thoughtful, constructive feedback provided. We are grateful for the time and effort you invested in the review process. We have revised the paper extensively to address all comments and believe the resulting manuscript is significantly clearer, better positioned, and more practically interpretable. Below we summarize the key changes made in response to your suggestions.

Clarification of theoretical assumptions and definitions.
We revised the null-space assumptions to explicitly use an
𝜀
ε-null space defined via an SVD cutoff, rather than assuming exact zero-variance directions. Theorem 1 (Variance–Leak) and related bounds were updated to include explicit
𝜀
ε-dependent residual terms, and corresponding thresholds were revised consistently. We also explicitly defined the model-induced distributions used in the analysis and clarified their role in Theorem 2. In addition, we introduced a dedicated “Notation and Abbreviations” subsection that defines activation matrices, null spaces, perturbations, and all probe quantities (NVL, SNL, FNC, BINA) in one place, ensuring that all symbols are defined before use.

Fisher Null-Conservation and robustness.
We relaxed the exact Fisher-silence assumption in Theorem 3 to an approximate condition
∥
𝐹
(
ℎ
)
𝑉
0
,
ℓ
∥
≤
𝛿
𝐹
∥F(h)V
0,ℓ
	​

∥≤δ
F
	​

. The theorem and proof were rewritten accordingly, showing that the KL conclusion holds up to a controlled second-order residual. We also clarified that Fisher Null-Conservation (FNC) can be measured empirically using standard Fisher approximations and added discussion interpreting regimes where NVL is small but FNC is large, guiding practitioners in using FNC as a diagnostic tool.

Empirical illustration and validation within a theory-first paper.
While the paper remains theory-focused, we added a minimal synthetic illustration as a targeted sanity check of a core theoretical prediction. The figure demonstrates that measured null leakage scales with
∑
𝑖
cos
⁡
2
𝜃
𝑖
∑
i
	​

cos
2
θ
i
	​

, as predicted by the Rank–Leak bound (Theorem 4), using controlled rank-
𝑟
r updates with prescribed principal angles. The caption and surrounding text were revised to make clear that this serves as qualitative validation rather than benchmarking.

Role of BINA and algorithmic clarity.
We clarified the role of the Bidirectional Null-Adversary (BINA), explicitly framing it as a diagnostic probe rather than a theorem-backed guarantee. A new paragraph explains how the BINA score
𝑆
B
I
N
A
S
BINA
	​

 should be interpreted as a practical stress test of functional sensitivity along null directions, complementary to NVL/SNL and FNC. We also clarified the distinction between model outputs
𝑓
(
ℎ
)
f(h) and the training loss
𝐿
(
ℎ
)
L(h) in Algorithm 1, and revised Algorithm 1—particularly Step 8—to explicitly describe the null-constrained adversarial update/evaluation being performed.

Positioning, related work, and scope.
We substantially rewrote the Related Work section to better position the framework within existing literature, organizing prior work around concrete theoretical gaps (e.g., dominant vs. silent subspaces, covariance vs. information geometry) and explicitly connecting these gaps to our results. We added a short “Theoretical framework and positioning” subsection to clarify how the individual theorems form a coherent framework. To clarify applicability to large language models, we added an explicit subsection grounding the theory in transformer-based LLMs, explaining how activation matrices, null spaces, and low-rank fine-tuning (e.g., LoRA) arise in practice. We also clarified the linear-algebraic scope of the framework while explicitly acknowledging nonlinear effects and positioning them as future work.

Finally, we ensured theorem numbering is now consecutive and consistent throughout the manuscript and resolved minor notation ambiguities.

We believe these revisions substantially improve the clarity, positioning, and accessibility of the manuscript, and we sincerely thank you for your constructive feedback, which materially strengthened the paper.

Sincerely,
The Authors

---

> ### Author Response · Authors · 2025-12-14
> **Question Regarding Uploading Revised Manuscript on OpenReview**
>
> Dear Editors,
>
> I hope this message finds you well. Thank you for coordinating the review of our manuscript and for the helpful feedback from the reviewers.
>
> We have completed the revisions in response to the reviews and would like to upload the updated manuscript and our response to reviewers. I wanted to confirm the correct procedure for submitting the revised version on OpenReview (e.g., whether it should be uploaded via the “Submit Revision” option on the existing submission, and whether any specific format is preferred for the response).
>
> Thank you very much for your guidance, and we appreciate your time and assistance.
>
> Sincerely, The Authors

---

### Author Response · Authors · 2025-12-16
**Follow-Up on Revised Manuscript**

Dear Reviewers,
We hope this message finds you well. We are writing to briefly follow up on the revised version of the manuscript, which we previously submitted in response to your comments.

When convenient, we would greatly appreciate any feedback you may have on whether the revisions adequately address the concerns raised, or if any additional clarification would be helpful.

Thank you again for your time and thoughtful engagement with our work.

Best Regards,
The Authors

---

> ### Author Response · Authors · 2025-12-25
> **Follow up**
>
> Dear Reviewers,
>
> I hope this message finds you well. I am writing to briefly follow up on our revised manuscript, which we previously submitted in response to your helpful comments.
>
> We have carefully addressed the feedback from all reviewers and made revisions aimed at improving clarity, definitions, and the positioning of the theoretical contributions. We would be very grateful for any further feedback you may have on the updated version when convenient.
>
> Thank you again for the time and effort you have invested in reviewing our work.
>
> Best regards,
> Authors

---

> ### Author Response · Authors · 2026-01-05
> **Happy New Year**
>
> Dear Editors,
>
> Happy New Year. We hope you are doing well.
>
> We are writing to kindly follow up on the status of our revised manuscript, for which we submitted the updated version addressing all reviewer comments a little over three months ago. We are grateful for the thorough and constructive review process so far and appreciate the time and effort invested by the editors and reviewers.
>
> As we enter the new year, we wanted to check whether there is an updated timeline for the completion of the review of the revised manuscript, or if any additional information is needed from our side to help move the process forward.
>
> Thank you again for your time and consideration. We look forward to your guidance on the next steps.
>
> Warm regards,
> Authors

---

### Decision · Action_Editor_DRC5 · 2026-01-11

**Recommendation:** Reject

**Additional Comments:**

Reviewer opinions were divided (two positive, one negative). I also share the concern that the connection to LLM monitoring is not established strongly enough in this submission, since empirical support is deferred elsewhere. The primary reasons for rejection are the bibliographic issues, which hinder verification and erode trust in the presentation.

**Audience:**

Yes

**Audience Explanation:**

Yes. The framing of drift through activation null spaces is conceptually interesting and could be relevant for monitoring and understanding drift in fine-tuned LLMs.

**Claims And Evidence:**

No

**Claims Explanation:**

As the reviewer recommendations were divided, I carefully read the submission. While several linear-algebra arguments appear plausible under the stated assumptions, key claims regarding practical implications in LLM monitoring are not supported by clear, verifiable evidence in the current manuscript.

Most importantly, there are bibliographic accuracy issues that prevent verification of the scholarly context and some supporting statements. For example, the submission cites arXiv:2410.17770, but the author list does not match the arXiv record. Additionally, I could not find the cited work “Jiaqi Feng, Jamie Smith, and Sarah Drews. Monitoring latent world states in large language models. (ICLR), 2024.” among the ICLR 2024 publications. These issues undermine confidence in the reliability of the related-work grounding.